# Ratiometric Fluorescent Sensor Based on Tb(III) Functionalized Metal-Organic Framework for Formic Acid

**DOI:** 10.3390/molecules27248702

**Published:** 2022-12-08

**Authors:** Chao-Wei Zhang, Mei-Ling Li, Yi-Duo Chen, Qi Zhou, Wei-Ting Yang

**Affiliations:** Key Laboratory of Advanced Materials of Tropical Island Resources, Ministry of Education, School of Chemical Engineering and Technology, Hainan University, Haikou 570228, China

**Keywords:** fluorescent metal-organic frameworks, lanthanide ions, ratiometric fluorescent sensor, formic acid

## Abstract

Formic acid is a common chemical raw material, the effective detection of which is of importance to food safety and environmental quality. In this work, the lanthanide functionalized dual-emission metal-organic framework (TH25) was prepared as a ratiometric fluorescent sensor for formic acid. This ratiometric sensor has a good detection performance with high selectivity, sensitivity, and reproducibility. Together with a low limit of detection of 2.1 ppm, these characters promise the ability to sense at low levels as well as a practical detection ability. This work provides ideas for the design and synthesis of effective chemical sensors for organic acids.

## 1. Introduction

As the simplest organic acid, formic acid (FA) is widely used in pesticides, leather, medicine, rubber, chemical raw materials, and other industries [1,2,3,4,5]. The use of FA in feed additives has an excellent acidification effect [6,7,8]. Thus the development and utilization of FA in feed additives have been increased gradually, and the application, scope, and dosage of FA have also increased. However, excessive exposure or inhalation of FA can cause a series of serious symptoms, such as skin diseases, bronchitis, chemical pneumonia, and even death [9,10]. As a result, the precise detection of FA is very important from the perspective of food safety and human health. So far, the technologies of FA detection mainly include chromatography [11,12,13] and electrochemical methods [14,15,16,17,18,19], however, both are cumbersome and expensive to operate. Recently, fluorescence detection has received increasing attention due to the advantages of low cost, easy operation, and high sensitivity. This technique can quantitatively or qualitatively analyze the fluorescence signal changes caused by the interaction between the fluorescent material and the analytes [20,21,22]. So far, various materials such as organic molecules and carbon dots have been used for the sensing of FA [23,24,25,26], however, most of the studies are based on the detection of gaseous FA. Therefore, the research for advanced fluorescent materials suitable for FA detection in a liquid-phase environment remains a great challenge.

Fluorescent metal-organic frameworks (FMOFs) have been widely used for the chemical sensing of various target analytes, such as organic small molecules [27,28,29], gases [30,31], and ions [32,33] due to their tunable structures and abundant fluorescent components. For example, Han et al. [34] constructed two Eu-based FMOF sensors that can be used for the rapid and selective detection of uric acid and adenosine triphosphate. However, FMOFs based on a single emission are susceptible to the interference of the external environment and show a limited detection accuracy [35]. The self-calibration strategy can overcome these shortcomings effectively and improve the stability of the sensor [36]. Furthermore, FMOFs featuring dual/multiple emissions have shown potential applications in ratiometric fluorescent sensing with high accuracy, anti-interference, and sensitivity [37,38,39,40]. Owing to the porous character [41] and the abundant active centers in the structures, FMOFs can be functionalized with the intrinsic emission of Ln^3+^ ions, organic dyes, or other fluorescent guests by post-modification. Furthermore, such functionalized FMOFs can be developed as advanced ratiometric sensing materials for a variety of analytes through monitoring the fluorescence intensity ratios [35,36,42,43,44]. For example, Sha et al. [45] reported a Zn-MOF with uncoordinated carboxylates, which provided binding sites for Eu^3+^ ions to form a dual-responsive FMOF sensor. The sensor realized the ratiometric sensing of arginine with the fluorescence color changing from blue to red. Despite many FMOF sensors having been reported, ratiometric FMOFs have never been used to detect FA.

In our previous work [46], it has been demonstrated that a Zn-based MOF (HNU-25 with the formula of {[Zn_2_(L)(tta)_2_]·3H_2_O·DMF·2Me_2_NH_2_^+^}, H_4_L is the abbreviation of 5-[bis(4-carboxybenzyl) amino] isophthalic acid, tta is the abbreviation of 1-H tetrazolium, and DMF is *N*,*N*-dimethylformamide as the skeleton can sensitize the characteristic fluorescence of Tb^3+^ ions. Herein, a Tb(III) functionalized dual-emission MOF (TH25) was constructed by introducing Tb^3+^ into HNU-25 through a host-guest interaction (Figure 1). It was found that the addition of FA destroyed the coordination between the Tb^3+^ ion and the framework with the fluorescence of the composite changing from green back to blue.

## 2. Results and Discussion

HNU-25 was prepared based on our previous report [46], which is a three-dimensional anionic framework with one-dimensional channels assembled by Zn^2+^ ions, L, and tta ligands (Figure 1). The powder X-ray diffraction (PXRD) pattern of the obtained colorless crystal is consistent with that simulated by a single crystal structure, indicating that HNU-25 of pure phase has been successfully synthesized (Figure 2a). Notably, the backbone of HNU-25 contains a large number of Lewis acid sites of carboxylate O and tetrazolium N atoms that are not involved in coordination; therefore they can further coordinate with lanthanide ions (Ln^3+^) to sensitize the characteristic luminescence of Ln^3+^, thus achieving a change from single-emission to dual-emission fluorescence. Based on this, we realized the sensing of Tb^3+^ by HNU-25, and the fluorescence changed from blue (emission of HNU-25) to green (emission of Tb^3+^). To demonstrate the interaction between HNU-25 and Tb^3+^ ions, X-ray photoelectron spectroscopy (XPS) was performed. The characteristic peaks of 1276.95 eV and 1242.22 eV appear in the full spectrum of TH25 (Figure 2b and Appendix A), indicating that TH25 contains Tb species. In addition, it can be found that the O1s peaks at 531.99 eV (C-O-Zn) and 533.97 eV (C=O) in HNU-25 shift to 531.71 eV and 533.43 eV, respectively. The N1s peak at 400.44 eV (C-N) in HNU-25 shifts to 400.80 eV after the addition of Tb^3+^ ions (Figure 2c,d), which can be attributed to the weak coordination between the Tb^3+^ ion and the carboxylate O as well as tetrazolium N atoms in the ligands.

Photoluminescence studies revealed that HNU-25 exhibited a strong emission band centered at 414 nm when excited at 365 nm (Figure 3), which can be attributed to the emission of the H_4_L ligand. While TH25 shows additional emissions at 489, 545, 585, and 620 nm corresponding to emissions of the ^5^D_4_ → ^7^F_J_ (J = 6, 5, 4, 3) transition → of Tb^3+^ ions. In addition, the fluorescence emission intensity of TH25 at 414 nm is significantly weaker than HNU-25, indicating that the interaction between the Tb^3+^ ion and the active sites in the structure can significantly enhance the energy transfer efficiency from the ligands to Tb^3+^ ions. Thus, TH25 exhibits a dual-emission fluorescence, which has the potential to be a ratiometric fluorescence probe.

According to the reported literature, we found that organic acids and bases can affect the binding of carboxylate groups to Ln^3+^ ions [47,48]. On this basis, a ratiometric fluorescence probe for a specific organic acid based on TH25 was developed. To investigate the sensing performance of TH25 for organic acids, the suspension of TH25 was added to a DMF solution containing different organic acids (FA, lactic acid (LA), citric acid (CA), acetic acid (AA), and oleic acid (OA)), and the fluorescence spectra were measured after mixing for 30 s. As shown in Figure 4a, the fluorescence intensity of TH25 at 414 nm remarkably improves with the addition of other organic acids except for OA, while the fluorescence intensity at 545 nm is significantly decreased. The addition of OA results in a decreasing trend of fluorescence intensity at both 414 nm and 545 nm. Meanwhile, the I_414_/I_545_ values of TH25 in the different organic acids can be used to represent its sensing performance. Figure 4b shows that the value of I_414_/I_545_ changes most significantly after the addition of FA, indicating that TH25 has the highest selectivity and sensitivity to FA.

The rapid response of TH25 to FA led us to further explore its detection sensitivity. TH25 was added to different concentrations of an FA solution and stabilized for 30 s before testing its fluorescence spectrum. As shown in Figure 5a, when FA concentration was gradually increased to 750 ppm, the fluorescence intensity at 414 nm gradually increased, and that at 545 nm gradually decreased. There was a good linear relationship between FA concentration and the fluorescence ratio of I_414_/I_545_ (Figure 5b). When the concentration of FA is in the range of 0–750 ppm, the linear relationship can be fitted by the equation of I_414_/I_545_ = 0.00109*C*_FA_ + 0.22278 (*R*^2^ = 0.9968). According to the equation of LOD = 3*δ*/*k* (LOD: limit of detection, *δ*: the standard deviation of blank parallel measurement (repeated more than 10 times), *k*: the slope of the fitted line, 0.00109, the specific calculation process is shown in supporting information), the LOD of TH25 for FA is 2.1 ppm. Compared with other existing sensors of FA, TH25 is the only fluorescent sensor for liquid FA, which is not only simple to operate but also exhibits a relatively low LOD (Table 1). In summary, TH25 is proved to be a good ratiometric fluorescent sensor for FA.

In the practical application of sensors, aside from sensitivity, repeatability, and anti-interference are also important performance evaluation indexes. As shown in Figure 6a, TH25 can be reused for FA detection after the treatment of an organic base, and it can still maintain a good performance after five cycles. The PXRD pattern after five cycles is basically consistent with the synthesized TH25, indicating that the TH25 sensor has good stability for FA sensing (Figure 2a). At the same time, the anti-interference property of TH25 was tested with the other coexisting components including ZnO, CuSO_4_, D-glucose, trypsin, L-histidine, L-cysteine, L-sarcosine, tyrosine, L-alanine, and L-glutamic acid, which are usually present in actual feeds. The fluorescence ratio of TH25 (I_414_/I_545_) to FA does not change significantly after adding these substances (Figure 6b). The above results indicate that TH25 can be used as the actual ratiometric fluorescence sensor for FA.

In general, the amount of FA added to pig and livestock feeds as a pH-reducing agent should not exceed 1.2% of the finished feed [6]. Otherwise, the growth cycle of livestock will be affected. To test the feasibility of the TH25 sensor in a practical application, different concentrations of FA (0, 25, 50, and 100 ppm) were added dropwise to equal amounts of feed (1 g of feed was used as the test condition). The concentration of FA was calculated from the established standard curve. As listed in Table 2, the relative standard deviations (RSDs) of the detection results are below 1.7%, and the spiked recoveries of FA in the feed are in the range of 95.8–109%, which indicates that the sensor can be used for the detection of FA in feed. These results demonstrate the application potential of TH25 in monitoring FA in real samples.

The sensing mechanism of FA by using TH25 was also studied. As shown in Figure 2a, the PXRD patterns of TH25 before and after FA sensing are basically consistent, and the infrared spectroscopy is almost unchanged (Appendix A), indicating that it is not due to structural collapse. XPS results show that the strength of Tb weakens after the introduction of FA, and the O 1s peaks at 531.71 eV and 533.43 eV shift to 531.79 eV and 533.73 eV, respectively (Figure 2c), which is due to the fact that FA destroys the coordination between carboxyl ate O and Tb^3+^ ion, affecting the energy transfer process between HNU-25 and Tb^3+^ ions to realize sensing. In addition, both the fluorescence intensity of TH25 at 414 nm and 545 nm can be recovered by adding triethylamine to the sensing system, which further proves the sensing mechanism. Based on the above studies, the sensing mechanism can be ascribed to the presence of FA causing some Tb^3+^ to be removed from its original binding site to change the fluorescence signal, which makes TH25 a promising sensor for FA sensing (Figure 1).

## 3. Materials and Methods

Preparation of HNU-25: A mixture of Zn (CH_3_COO)_2_ (0.0183 g, 0.1 mmol), H_4_L (0.0225 g, 0.05 mmol) and tta (0.014 g, 0.2 mmol) was dissolved in a glass vial (25 mL) of 10 mL DMF with 1 mL of deionized water. Then a drop of CH_3_COOH (99.5%, aq.) was added. After that, it was kept at 85 °C for 3 days and cooled at room temperature to obtain colorless crystals.

Preparation of TH25: 50 mg of fully ground HNU-25 was added to 10 mL of Tb^3+^ solution (DMF) with a concentration of 500 ppm. Then, TH25 was obtained by centrifugal drying. 

Determination of different organic acids: At room temperature, 5 mg TH25 was added to 750 ppm of different organic acid solutions (DMF), and the fluorescence data were measured after 30 s.

Determination of FA in solution: At room temperature, 10 mg TH25 was added to FA solution (DMF) of different concentrations, and fluorescence data were measured after 30 s.

Determination of FA in feed: Feed with different FA supplemental levels (M_FA_/M_feed_ = 0, 0.6%, 1.2%, and 2.4%) was immersed in 10 mL DMF solution for 1 h to obtain the stock solution. Then 4 mg TH25 was added to the stock solution with different concentrations, and the fluorescence data were measured after 30 s.

## 4. Conclusions

In conclusion, a Tb-functionalized dual-emission FMOF sensor was constructed by the post-synthesis method for ratiometric sensing of FA. The introduction of Tb^3+^ ions can not only modify the fluorescence characteristics of MOF but also act as a responsive site for FA. As a ratiometric fluorescent sensor, TH25 shows the advantages of rapid response, high selectivity, good repeatability with maintaining detection performance after five cycles, and a low detection limit of 2.1 ppm. Furthermore, TH25 can achieve the detection of FA in the liquid phase as well as in the actual feed samples. This work not only provides a convenient strategy for FA sensing but also opens a path for MOF-based ratiometric fluorescent sensors.

## Figures and Tables

**Figure 1 molecules-27-08702-f001:**
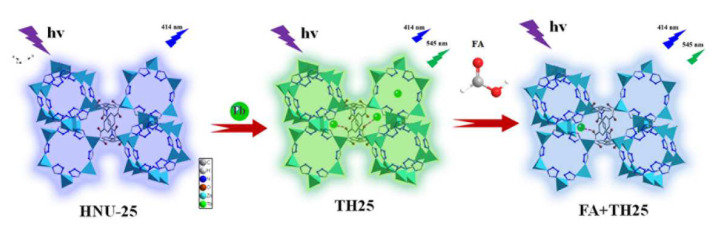
Schematic diagram showing the design and detection process of TH25 as fluorescence probe for FA.

**Figure 2 molecules-27-08702-f002:**
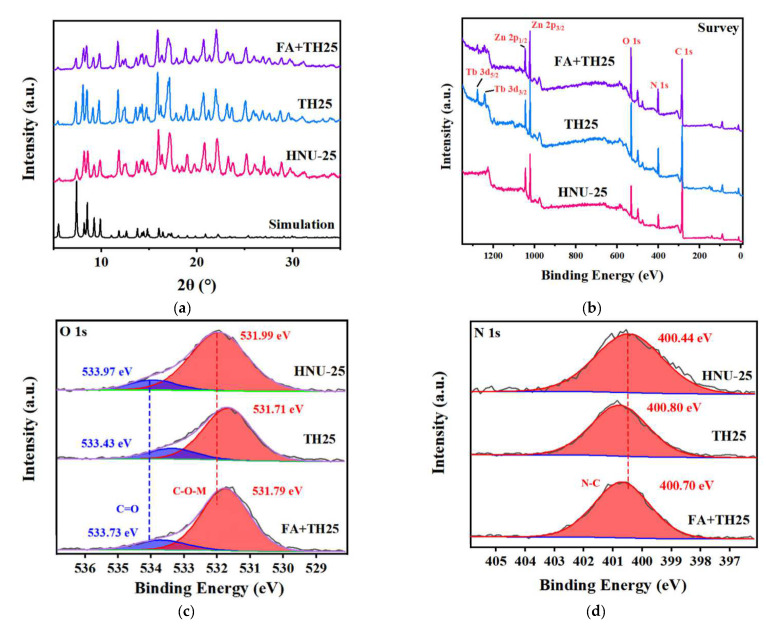
(**a**) PXRD patterns of simulation, HNU-25, TH25, and FA + TH25; (**b**) XPS survey spectra of HNU-25, TH25, and TH25 + FA; high-resolution XPS spectra of O 1s (**c**) N 1s (**d**).

**Figure 3 molecules-27-08702-f003:**
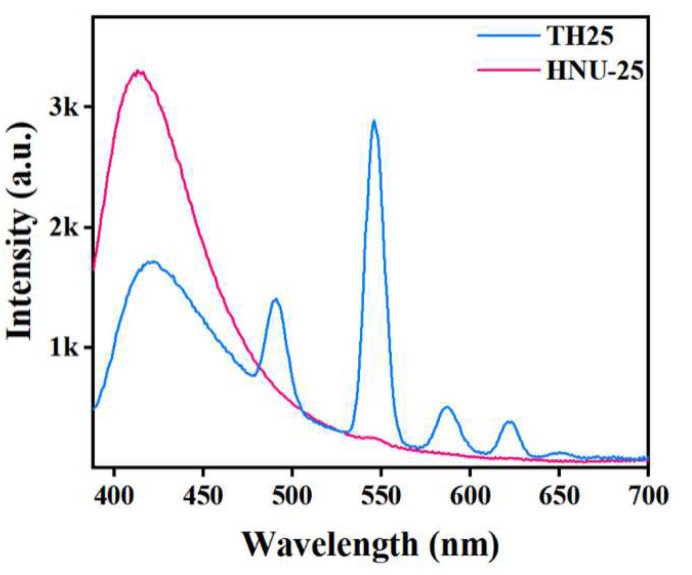
Solid-state fluorescence spectra of HNU-25 and TH25 (*λ*_ex_ = 365 nm).

**Figure 4 molecules-27-08702-f004:**
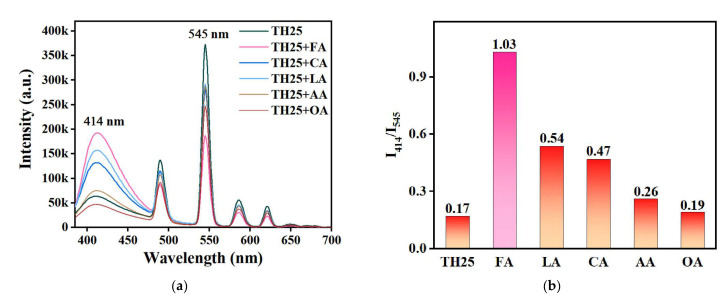
(**a**) Fluorescence spectra of TH25 with the addition of different organic acids (*λ*_ex_ = 365 nm); (**b**) Fluorescence ratio of emission intensity at 414 nm and 545 nm in TH25 upon addition of different organic acids.

**Figure 5 molecules-27-08702-f005:**
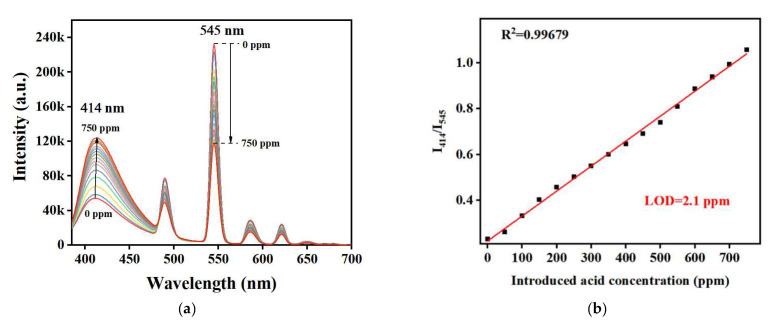
(**a**) Fluorescence spectra of TH25 with an increase in the concentration of FA (*λ*_ex_ = 365 nm); (**b**) Fluorescence ratio of TH25 (measured at 414 nm and 545 nm) is fitted to the linear relationship of FA concentration.

**Figure 6 molecules-27-08702-f006:**
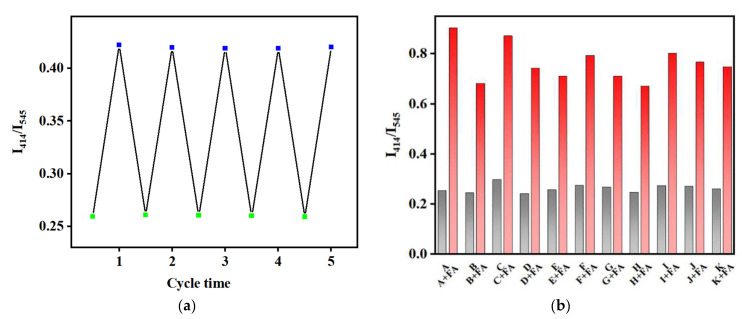
(**a**). Proportional fluorescence intensity of TH25 placed in DMF solution containing FA (blue, 500 ppm) and triethylamine (green, 500 ppm); (**b**). Interference of different substances with FA detection (fluorescence ratio measured at 414 nm and 545 nm of TH25 for FA (A) in presence of ZnO (B), CuSO_4_ (C), D-glucose (D), Trypsin I, L-Histidine (F), L-Cysteine (G), L-Alanine (H), L-Glutamic (I), L-Sarcosine (J), Tyrosine (K)).

**Table 1 molecules-27-08702-t001:** Comparison between TH25 and Reported Methods for Detecting FA.

Material	Phase State of FA	Detection Method	LOD (ppm)	Ref
TH25	Liquid	Ratio fluorescence	2.1	this work
D-A fluorophore	Gas		8.7 × 10^−2^	[23]
Carbon dots			150	[24]
MOF-802	Liquid	Electrochemical analysis	14.7	[16]
Cobalt (II)-MOF 1			35.0	[17]
ZZU-1			552	[19]
-	Liquid	Microextraction—liquid chromatography	3.0 × 10^−4^	[11]

**Table 2 molecules-27-08702-t002:** Detection of FA in Feed with TH25 Sensor (*n* = 3).

Added *^a^* (ppm)	Detected (ppm)	RSD (%)	Recovery (%)
0	0	-	-
25	24.4	0.03	97.6
50	47.9	0.05	95.8
100	109	1.66	109

*^a^* Added FA 25 ppm, 50 ppm, and 100 ppm means M_FA_/M_feed_ = 0.6%, 1.2%, and 2.4%.

## Data Availability

Not applicable.

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
