# Peer review of "Ratiometric Fluorescent Sensor Based on Tb(III) Functionalized Metal-Organic Framework for Formic Acid"

_molecules, 2022, doi:10.3390/molecules27248702_

Round 1
Reviewer 1 Report
The manuscript entitled " Ratiometric fluorescent sensor based on Tb(III) functionalized metal-organic framework for formic acid". The reported values are acceptable due to broadly explanation. Authors reported synthesis of HNU-25. A Part of these authors also mentioned the Photoluminescence studies of the compounds. The conclusions drawn from the studies are clearly stated and are reasonable. This is an interesting article addressing an important issue and very well presented, and is within the scope of this journal. I recommend its publication once the minor issues are addressed.
1. In its current state, the level of English throughout the manuscripet does not meet desired standard. There are a number of grammatical errors and instances of badly worded/constructed sentences. Please check the manuscript, refine the language.
2. The introduction still does not properly highlight the purpose of the work. The authors should add some relevant information about the application of fluorescein in the introduction. Following references authors should introduced. Journal of Photochemistry and Photobiology A: Chemistry 406 (2021) 113022, Journal of Molecular Liquid 328 (2021) 115407. J Fluoresc (2016) 26:937–947, Journal of Molecular Liquids 243(2017) 85-90. Journal of Solution Chemistry (2018) 47:1711–1724, Journal of Coordination Chemistry, 73 (2020) 2987-3002. Journal of Luminescence 137 (2013) 6-14. Journal of Molecular Liquids 216 (2016) 423-428.
3. Author should explain the effect of alcoholic solvent
4. Conclusion is not appropriate. It should summarize major findings.
Reviewer 2 Report
Inn this manuscript, Yang and coworkers are reporting an interesting Tb(III)-MOF-based fluorescence sensing method for sensing method to detect formic acid. The overall layout of the manuscript seems to be reasonable. However, I suggest authors to answer following concerns in the revision.
(1). What is the significance of this develop method over the existing methods? Can authors explain this briefly?
(2). LOD calculations must be provided in the supporting information and this value must be compared with other reported detection methods from the literature.
(3). Can authors provide a "chemical scheme" clearly explaining the sensing mechanism?
(4). Can authors comment on the signal to noise ratio in this method?
(5). What is the binding stoichiometry?
(6). There are several grammatical and textual errors may need to be revised throughout the manuscript.
